# New approach for FIB-preparation of atom probe specimens for aluminum alloys

**L. Lilensten**[1¤]*, **B. Gault**[1,2]

**1** Department of Microstructure Physics and Alloy Design, Max-Planck-Institut für Eisenforschung GmbH, Düsseldorf, Germany, **2** Department of Materials, Imperial College London, Royal School of Mine, London, United Kingdom

¤ Current address: Institut de Recherche de Chimie Paris, PSL Research University, Chimie ParisTech, CNRS UMR 8247, Paris, France
* lola.lilensten@chimieparistech.psl.eu, l.lilensten@mpie.de

## Abstract

Site-specific atom probe tomography (APT) from aluminum alloys has been limited by sample preparation issues. Indeed, Ga, which is conventionally used in focused-ion beam (FIB) preparations, has a high affinity for Al grain boundaries and causes their embrittlement. This leads to high concentrations of Ga at grain boundaries after specimen preparation, unreliable compositional analyses and low specimen yield. Here, to tackle this problem, we propose to use cryo-FIB for APT specimen preparation specifically from grain boundaries in a commercial Al-alloy. We demonstrate how this setup, easily implementable on conventional Ga-FIB instruments, is efficient to prevent Ga diffusion to grain boundaries. Specimens were prepared at room temperature and at cryogenic temperature (below approx. 90K) are compared, and we confirm that at room temperature, a compositional enrichment above 15 at.% of Ga is found at the grain boundary, whereas no enrichment could be detected for the cryo-prepared sample. We propose that this is due to the decrease of the diffusion rate of Ga at low temperature. The present results could have a high impact on the understanding of aluminum and Al-alloys.

## Introduction

Penetration of Ga at grain boundaries in pure-Al and Al-based alloys is a well-known problem. It indeed has highly detrimental consequences, such as liquid metal embrittlement, causing inter-granular brittle fracture of these materials [1]. This creates issues when it comes down to analyzing aluminum samples by transmission electron microscopy and atom probe tomography (APT), for which specimen preparation is increasingly performed via Ga-beam based focused ion beam (FIB) milling. APT allows to investigate the chemical effects in materials at the atomic scale. It is of high relevance when it comes to the analysis of impurities and solute segregation at grain boundaries for example, since they can, even in trace concentrations, have tremendous effects on grain growth, mechanical properties, or possible grain boundary precipitation for instance [2–4]. APT is thus an appointed technique. However, high quality analyses require a high quality of sample preparation, in order to avoid introducing artefacts in the

**Data Availability Statement:** All relevant data are within the paper and its Supporting Information files. All APT files are available in figshare (https://figshare.com/s/f5b51f3839cd0d3564dc).

**Funding:** L.L. thank the Alexander von Humboldt Foundation for the financial support. L.L. and B.G.

are grateful for funding from the ERC-CoG SHINE – 771602. The funders provided support in the form of salaries for authors LL and BG, but did not have any additional role in the study design, data collection and analysis, decision to publish, or preparation of the manuscript. The specific roles of these authors are articulated in the 'author contributions' section.

**Competing interests:** L.L. thank the Alexander von Humboldt Foundation for the financial support. L.L. and B.G. are grateful for funding from the ERC-CoG SHINE – 771602. The funders provided support in the form of salaries for authors LL and BG, but did not have any additional role in the study design, data collection and analysis, decision to publish, or preparation of the manuscript. The specific roles of these authors are articulated in the 'author contributions' section. The commercial affiliation to the Max Planck Institut für Eisenforschung GmbH does not alter our adherence to PLOS ONE policies on sharing data and materials.

specimen. Nowadays, APT sample preparation benefits from advanced protocols using FIB—scanning electron microscopes (SEM), that allow to perform site-specific lift-outs, and target desired features, such as grain boundaries or precipitates, with a high efficiency [5]. The lifted out specimens are then sharpened in the shape of a needle with an end radius in the range of 50–150nm. Conventional FIB machines use a Ga-ion source for the milling part. Although this technique is adequate for the study of most materials, it cannot be easily applied for the study of aluminum and its alloys: analyses of the grain boundaries composition, and investigation of potential Ga traces there, but also at other defects such as dislocations, is affected by Ga-ions implantation and diffusion due to the FIB preparation [6–10]. Existing alternatives include electropolishing, but this technique is not site specific, or FIB preparation using a Xe+ ion beam [10–12]. The last technique has shown a great efficiency, but such instruments are not widely available.

Accounting for the fact that Ga diffusion at the grain boundary happens when Ga is in its liquid form, we propose in the present study to investigate the Ga-ion FIB preparation for grain boundaries in Al-alloys samples at cryogenic temperatures. The same principle recently proved successful in limiting hydrogen ingress during FIB-specimen preparation of commercially pure Ti [13]. Our results show that going down to cryogenic temperature allows to remove the grain boundary diffusion of Ga, and therefore to produce clean specimen for APT analyses.

## Materials and methods

The APT specimens were taken from the mechanically polished surface of a commercial 6016 aluminum alloy. First, a lamella at a grain boundary was prepared following a site-specific lift-out procedure described elsewhere using a commercial microtip coupon as a support [5]. Sharpening of the APT specimen was performed on a FEI Helios 600 dual-beam scanning electron microscope/focused-ion beam (SEM/FIB) using a Ga ion source. Two different setups were used: first, sharpening was done using a conventional FIB stage, at room temperature (293K). Second, sharpening was done with the coupon mounted on a commercial cryo stage (Gatan C1001). A home-designed holder was used to hold the mounted coupon at 52° from the stage, i.e. directly aligned with the Ga-beam. The stage was cooled down by a continuous $N_2$ flow, ensuring a stable temperature of 82K during milling. After sharpening, the specimens were warmed up to room temperature within approx. 15min. In both cases, an acceleration voltage of 30kV was used, and the currents used during sharpening were 0.28nA for an inner diameter above or equal to 1 micron, 93pA for diameters below 1 micron and down to 0.5 micron, and 48pA for diameters below 0.5 micron. A final cleaning step at low voltage (5kV, 47pA) was finally performed to remove the material likely affected by implantation damage. Specimen were then transferred in the air at room temperature to the Cameca LEAP 5000XR. Analyses were performed at 80K, in voltage pulse mode, with a pulse fraction of 20%, a pulse rate between 50kHz and 100kHz, and a target detection rate of 0.5%. The temperature for the analysis was chosen to maximize the yield [14]. Data analyses were performed with the IVAS software. The reconstructions were calibrated using crystallographic poles using the protocol outlined in [15]. Detector maps were recalculated using the information extracted from the epos using routines coded in Matlab.

## Results

First, a specimen prepared at room temperature is analyzed. The analyzed volume displays a region that contains a high density of Ga, almost perpendicular to the needle axis, as shown in the reconstruction displayed in Fig 1A. Only part of the reconstruction is displayed here. This

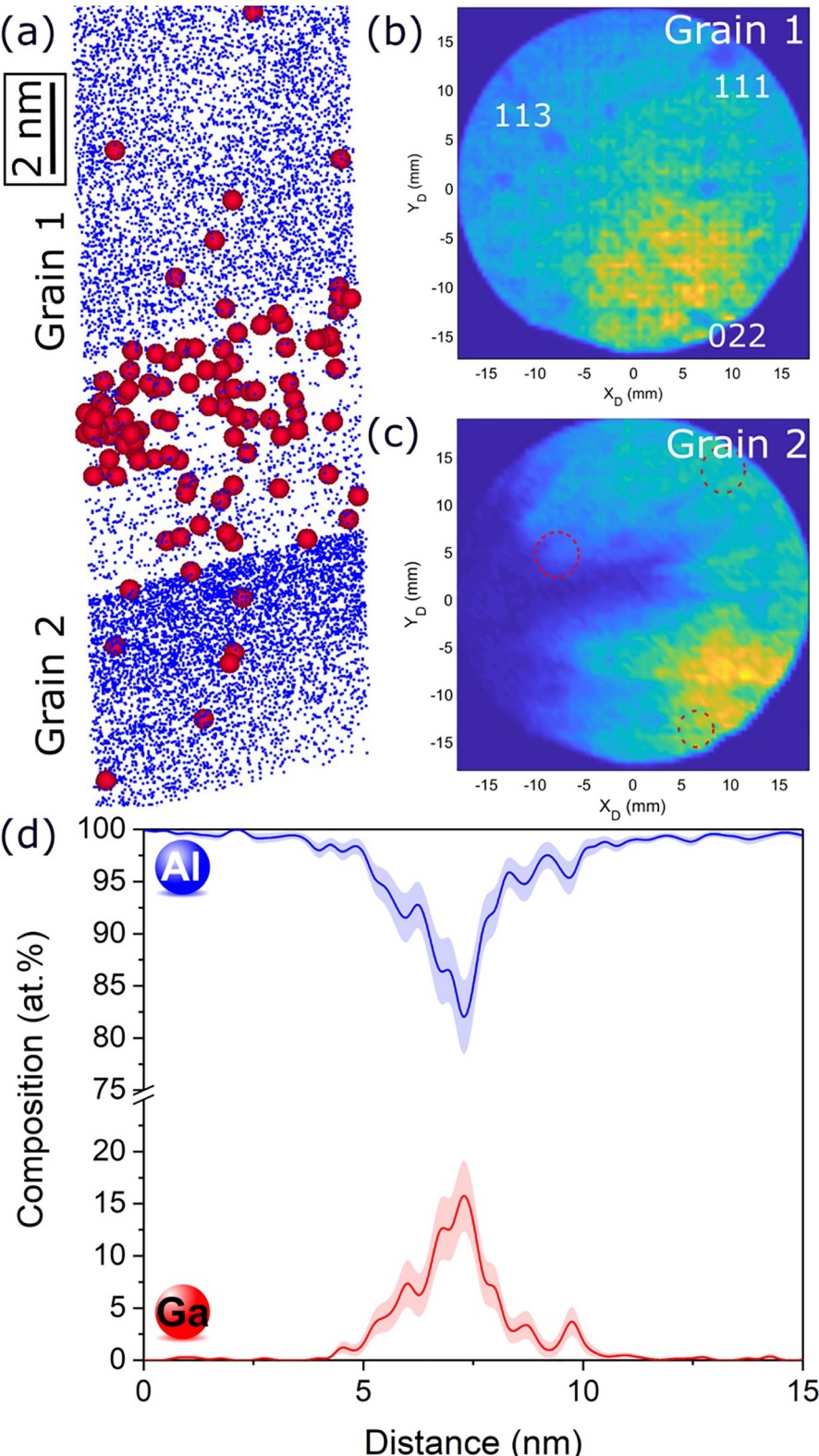

**Fig 1. Atom probe analysis of a grain boundary in a sample prepared at room temperature.** (a) Part of the APT reconstruction of the specimen prepared by conventional FIB at room temperature. Atomic planes are visualized in grain 1 (top), and disappear in grain 2. Al atoms are represented as blue dots, and Ga atoms as red circles. (b) XY evaporation histogram of the detector showing crystallographic poles for grain 1. (c) XY evaporation histogram of the

detector showing crystallographic poles for grain 2. The poles of grain 1 are reported on this histogram as red dashed circles for an easier readability. (d) Composition profile for Al and Ga across the grain boundary. Error bars are shown as lines filled with colour and correspond to the 2σ counting error.

region has been calibrated using the plane interspacing distances, and Al planes are indeed evidenced in the upper part of the reconstruction. Below the Ga-rich region (dense band of red points, corresponding to Ga ions), one can see that the reconstruction changes: the planes are no longer visible. The point density appears higher, owing to a different evaporation field which can be related to the change in crystallographic orientation as well as the change in the field evaporation behavior associated to the high Ga content [16]. Detector maps are displayed for the upper part of the reconstruction in Fig 1B, and for the lower part in Fig 1C. For an easier readability, the poles observed for the upper part are reported on the lower evaporation histogram as red dashed circles. Although fracture of the specimen occurred soon after the Ga dense region, leading to an evaporation histogram of lower quality for the bottom part, some poles are still observed in Fig 1C. As highlighted by the red dashed circles, it appears that there is a shift of the position of the main poles in Fig 1B. This is an indication that the upper and lower parts, on either side of the Ga-rich region, have different crystallographic orientation, and this information suggest that the dense Ga region corresponds to a grain boundary. This region also seems to have a lower density when looking at the Al in Fig 1A. This is likely related to the field evaporation behavior that is modified by the combination of a grain boundary [17] and the local presence of the high amount of Ga that seems to exhibit a low evaporation field and hence evaporated in a burst and led to significant distortions as can be observed for very low evaporation fields particles [18]. Specimen failure in the vicinity or at the grain boundary itself occurred in several datasets, and can likely be attributed to Ga-induced embrittlement. A composition profile is plotted across the grain boundary (for the region displayed in Fig 1A) and is given in Fig 1C. It shows a clear Ga enrichment at the grain boundary, reaching values above 15 at.%, along with a depletion of Al.

Such an accumulation of Ga has previously been reported to pertaining to microstructural features such as dislocations or grain boundaries [7,10,19]. Here, our results suggest that the characterized feature corresponds to the grain boundary that we targeted during the preparation of the specimen. This explains the drastic planar Ga-enrichment. The fracture of the specimen soon after is also in line with embrittlement of grain boundaries caused by the Ga-indiffusion.

A specimen prepared by cryo-Ga FIB was then analyzed. A part of the reconstruction, featuring two grains, is presented in Fig 2A. Planes are evidenced for the upper grain (grain 1), and then disappear approximately at the pink dashed line (guide for the eyes), which corresponds to the orientation difference between grain 1 and grain 2. To ensure that two grains are indeed analyzed in the reconstruction, crystallographic analyses were performed. The APT dataset is sliced into bins of a million ions, and for each slice, the corresponding detector hit map was plotted. The results, shown in Fig 2B and 2C, evidence a change of crystallographic orientation, confirming the presence of two grains and subsequently of a grain boundary. A composition profile across the grain boundary, going from grain 1 to grain 2 (in Fig 2A) is then calculated. This time, no Ga segregation is evidenced there.

## Discussion

The diffusion of Ga at Al grain boundaries is a well-known problem that has dramatic consequences on Al-alloys mechanical properties [1]. Most studies suggest that the decrease of the interfacial energy, through the formation of a Ga film of a few atomic layers in thickness, is the

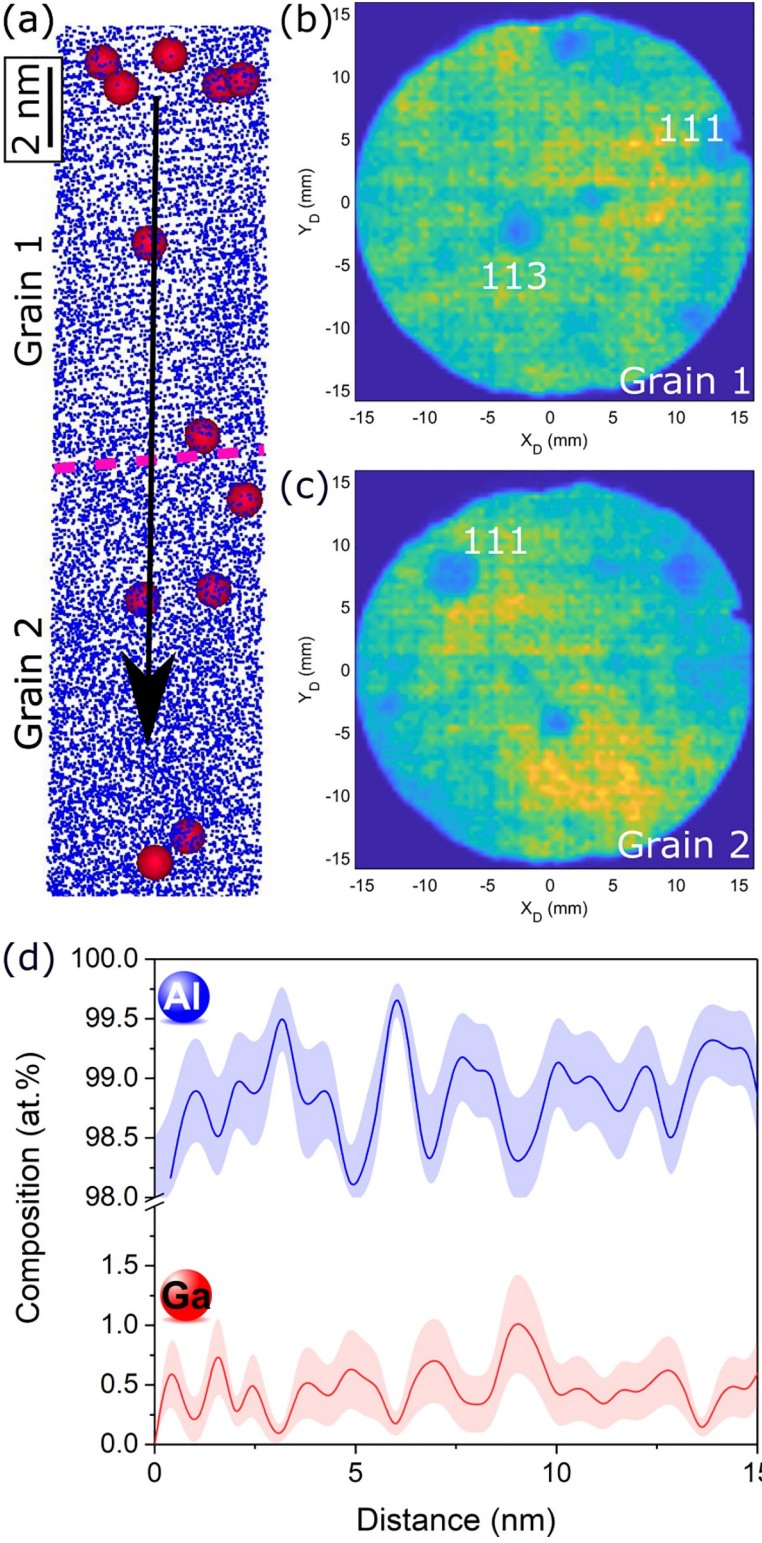

**Fig 2. Atom probe analysis of a grain boundary in a sample prepared at cryogenic temperature.** (a) part of the APT reconstruction of the specimen prepared by cryo-FIB. Atomic planes are evidenced above the pink dashed line, corresponding to grain 1, and disappear below the line, corresponding to grain 2. Al atoms are represented as blue dots, and Ga atoms as red circles. (b) and (c) XY evaporation histogram of the detector showing crystallographic poles for grain 1 and grain 2, respectively. (d) Composition profile for Al and Ga across the grain boundary, along the black

arrow in (a). Error bars are shown as lines filled with colour and correspond to the 2σ counting error. No Ga segregation is observed.

driving force, but other hypotheses, such as the formation of intermetallic compounds, have also been proposed [20,21], and the exact nature of the atomistic mechanism behind liquid-metal embrittlement remains debated in the community. Therefore, based on the grain boundary nature (high-angle or low-angle grain boundary, coincidence site lattice (CSL) boundaries), but also the applied stress, the speed of penetration of Ga changes [21–25]. Based on an in situ TEM study at room temperature, the penetration speed of Ga at grain boundaries varies between 0.01 to 12.2 μm.s$^{-1}$ [22]. In this context though, the temperature was not measured but a reasonable estimate would be that the temperature under the beam is in the range of 298–323K, because of a possible temperature increase caused by the illumination by the electron beam. Although it is difficult to estimate the exact propagation speed of Ga along the grain boundary during room temperature milling, because of several unknown parameters such as the local stress or the nature of the grain boundary, as well as possible ion channeling, it is reasonable to consider that the entire grain boundary is affected by Ga during milling at room temperature. Indeed, milling times of well above a second are applied (typically in the order of a minute for the first steps, down to a few seconds in the last ones), and the initial sample before sharpening has a maximum width of about 3 μm, this value being reduced progressively as sharpening proceeds, leading to rather short diffusion distances to cover.

For the cryo-temperature experiment, the diffusion is much reduced. Indeed, based on the results of Peterson and Rothman, the diffusion coefficient of Ga in Al is $D(T) = 4.90.10^{-5}exp\left(-\frac{(122.34\pm0.59)10^3}{RT}\right)$, with the pre-exponential $D_0$ factor in m$^2$.s$^{-1}$ and the activation energy is expressed in J.mol$^{-1}$ [26]. Calculation of the diffusion coefficient at room temperature and at cryogenic temperature (experimental temperature of 82K) leads to D(RT) = 5.57.10$^{-27}$ m$^2$.s$^{-1}$ and D(cryo) = 5.70.10$^{-83}$ m$^2$s$^{-1}$, respectively. This leads to diffusion distances in the range of less than a nanometer per minute at room temperature inside the grains, which can translate into over a nm per minute along grain boundaries. With specimen's sizes in the range of only 100nm, it is possible that Ga ends up covering a significant fraction of a GB located within an atom probe specimen.

At cryogenic temperature, even though the diffusion at grain boundaries is faster than in a single crystal, the difference in diffusion length of tens of orders of magnitude tend to confirm that the diffusion is likely almost stopped, even at grain boundaries. These results are in perfect agreement with the experimental results, suggesting that for cryo-preparation, the Ga diffusion distance is smaller than the sample thickness removed during the next step of the sharpening process. The Ga level measured there (0.25–0.5 at.%) is still higher than what can be expected for the commercial alloy. It is believed to originate from the sample preparation, as evidenced in the S1 Fig, which shows a contamination of Ga at the surface of the specimen, and a decrease of this content as the evaporation proceeds (see composition profile). A mass spectrum also shows that one isotope is predominantly obtained, reinforcing the hypothesis that the measured Ga comes from the FIB preparation. This does not impede the result that Ga does not segregate at the grain boundary.

The schematic in Fig 3 summarizes the proposed consequences of room temperature and cryo preparation: Ga is implanted at the surface at each cut, with decreasing thickness due do the decreased current, but the milling provided by the new cut also removes the affected zone of the previous cut, hence cleaning the implantation. Diffusion is also observed at the GB. For cryo preparation, both Implantation and diffusion at GBs are reduced thanks to the low

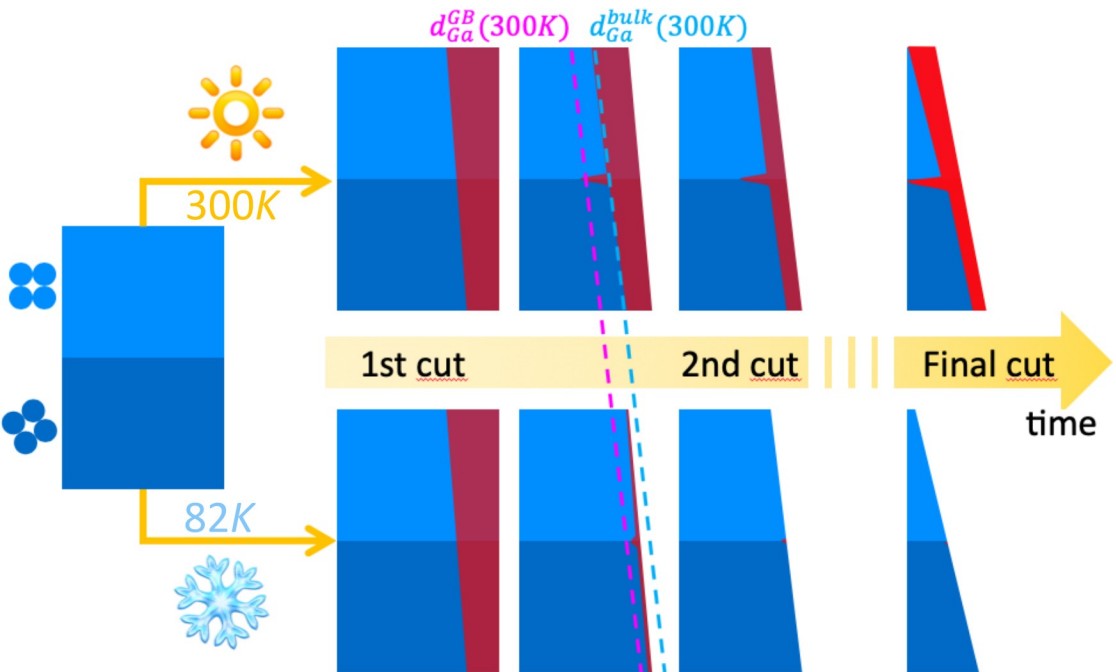

**Fig 3. Schematic of the Al-Ga interaction during APT sample preparation for the two preparation routes.** Schematic of the evolution of Ga implantation and diffusion at grain boundaries during sharpening of the APT sample, for conventional room temperature preparation (top) and for cryo preparation (bottom). The two grains are represented in light and dark blue, and the Ga affected zone in red.

temperature, enabling efficient removal of the affected surface, and production of a sample without specific Ga implantation at the GBs.

The fact that the Ga composition does not increase specifically at the grain boundary, although the specimen was transferred at room temperature, suggests that cryo-transfer of the specimen is not required. Even in the present case where Ga contamination at the near-surface region occurred, and led to Ga diffusion in the specimen in the room-temperature transfer, it did not reach a critical level affecting the grain boundary. Therefore, cryo preparation allows for reliable data analyses, thanks to a removal of Ga at grain boundaries during sample preparation.

## Conclusion

The present study proposes a new preparation route for APT samples of aluminum and its alloys. To tackle the diffusion at grain boundaries of gallium, originating from the FIB preparation and degrading the quality of the analysis and the data, a Ga-FIB cryo-preparation protocol is suggested. Comparative APT experiments on a grain boundary of a commercial 6016 aluminum alloy show that the gallium composition at the grain boundary, above 15 at.% in the case of a room temperature Ga-FIB preparation, is reduced close to a very low level (fluctuating between 0.25 and 0.5 at.%) which does not increase at the grain boundary in the case of a cryo-FIB preparation. This new protocol, easy to implement on existing Ga FIBs, could therefore enable a much more efficient and cheaper way of preparing Al specimen by FIB techniques, unlocking current technological limitations for a better understanding of aluminum and its alloys.

## Supporting information

**S1 Fig. Ga implantation during milling for the cryo-prepared specimen.** (a) Atomic reconstruction and (b) composition profile along the arrow in (a) for the Gallium showing a composition gradient from the top of the specimen to the bottom. (c) Corresponding mass spectrum of the dataset, that was cut at 100 Da (no peaks are observed at larger Da). The inset shows a zoom of the region between 64 and 74 Da, with the 69Ga+ peak and a much smaller 71Ga + peak. The little peak at 65 Da corresponds to Cu, which is a classical impurity found in low concentrations.
(PDF)

## Author Contributions

**Conceptualization:** L. Lilensten, B. Gault.

**Data curation:** L. Lilensten, B. Gault.

**Formal analysis:** L. Lilensten, B. Gault.

**Investigation:** L. Lilensten, B. Gault.

**Methodology:** L. Lilensten, B. Gault.

**Writing – original draft:** L. Lilensten.

**Writing – review & editing:** B. Gault.

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
