## [Decision Letter · Decision Letter 0]

2 Jan 2020

PONE-D-19-33571

New approach for FIB-preparation of atom probe specimens for aluminum alloy

PLOS ONE

Dear Dr Lilensten,

Thank you for submitting your manuscript to PLOS ONE. After careful consideration, we feel that it has merit but does not fully meet PLOS ONE’s publication criteria as it currently stands. Therefore, we invite you to submit a revised version of the manuscript that addresses the points raised during the review process.

We would appreciate receiving your revised manuscript by Feb 16 2020 11:59PM. To enhance the reproducibility of your results, we recommend that if applicable you deposit your laboratory protocols in protocols.io, where a protocol can be assigned its own identifier (DOI) such that it can be cited independently in the future. For instructions see: http://journals.plos.org/plosone/s/submission-guidelines#loc-laboratory-protocols

We look forward to receiving your revised manuscript.

Kind regards,

Hamid Reza Bakhsheshi-Rad

Academic Editor

PLOS ONE

Journal Requirements:

2. Please note that PLOS ONE publishing criteria require that experiments must be described in sufficient details for another researcher to reproduce the findings (http://journals.plos.org/plosone/s/criteria-for-publication#loc-3). In light of this please amend your methods section to include the source and manufacturer name of all the materials, chemicals and instrumentation used.

L.L. thank the Alexander von Humboldt Foundation for the financial support. L.L. and B.G. are grateful for funding from the ERC-CoG SHINE – 771602.

We note that one or more of the authors are employed by a commercial company: Max-Planck-Institut für Eisenforschung GmbH

6. Please amend either the title on the online submission form (via Edit Submission) or the title in the manuscript so that they are identical.

Additional Editor Comments:

I have completed the review of your manuscript and a summary is appended below. The reviewers recommend reconsideration of your paper following major revision. I invite you to resubmit your manuscript after addressing all reviewer comments.

Reviewers' comments:

Reviewer's Responses to Questions

**Comments to the Author**

1. Is the manuscript technically sound, and do the data support the conclusions?

Reviewer #1: Partly

Reviewer #2: Partly

2. Has the statistical analysis been performed appropriately and rigorously? 

Reviewer #1: Yes

Reviewer #2: N/A

3. Have the authors made all data underlying the findings in their manuscript fully available?

Reviewer #1: Yes

Reviewer #2: Yes

4. Is the manuscript presented in an intelligible fashion and written in standard English?

Reviewer #1: Yes

Reviewer #2: Yes

5. Review Comments to the Author

Reviewer #1: In this paper two atom probe Al alloy samples are compared, one prepared with a FIB at room temperature and one prepared with the FIB cooled to cryogenic conditions. This is usually accompanied with a large effort to do, especially when investigating grain boundaries where Ga is present, which is known to lead to bad yield in the atom probe experiment. It is stated that Ga has a smaller diffusion length at lower temperatures, which is also underlined with a calculation of the diffusion coefficient. And concluded that no enrichment for Ga at the grain boundary in the cryo prepared sample was obtained.

However, in general the paper would profit from a better representation, that in the room temperature FIB sample the second grain was already measured (as for Figure 2 b)). In general a larger concentration of Ga does not necessarily ensure the existence of a grain boundary. A visual representation of the grain boundary with an iso surface of the Ga distribution in the sample, or a Ga XY evaporation diagram, would be benefitial. Further the verification that the second grain was measured would be benefitial, either with showing loss of crystal information from grain to grain via SDMs or showing the different orientation with the XY evaporation diagram as in Figure 2 b), the reviewer can hardly see a difference in the shown ROI from grain 1 to grain 2 in Figure 1, apart from the low density region in between with increased Ga concentration, followed by a higher density region.

The behavior of Si at the boundary is described as increased at the boundary from almost 0% to 1.5 at.% the boundary, which is rather unexpected to the reviewer. Usually the Si content is in the order of 1 at.% in general in the alloy 6016. Further, Si is known to be seen to be present with a higher density in the reconstruction at the (111) pole. This pole is also seen on the XY evaporation diagram. In Figure 1 it is not indicated where the ROI was taken from, therefore if the (111) pole is presented, this could cause an increase of Si in the chosen ROI and potentially influence the evaluation of seen Si increase. However, the reviewer would rather expect that the boundary influences the behavior of Mg more, for both elements (Mg, Si) an evaluation as in Figure 1c would be good additional information. A scale for Figure 1a and Figure 2a would also be benefitial.

I cannot agree with the statement that "Ga is reduced close to 0 at.% (comparable to the level of background)" when comparing it to Figure 2d showing about 0.25 - 0.5 at.% Ga at average, which is clearly above usual background and in the order of the major alloying element concentrations (like Mg usually in the order of 0.4 at.% for 6016). Although Ga is present in industrial alloys, it only occurs in approx. two orders of magnitude less, which means that if there are Ga peaks present at this concentrations they are from FIB-implanted Ga.

The specimen was transfered at room temperature to the atom probe, which thwarths somehow the argument that the cooling prevented Ga diffusion to the boundary / at the boundary.

Some minor comments:

- page 3: not "tension of 30 keV", "acceleration voltage of 30 keV"

- experiments are examined at a rather "high" temperature of 80

- indicating the major poles of Figure 1b and Figure 2b,c would increase readability

- There seems to be a lower density region where high amounts of Ga are present (Figure 1a), which is somehow misleadingly formulated at the main text.

Reviewer #2: The work compares the preparation of APT samples using FIB at RT and cryogenic conditions. Ga induced via RT samples preparation often limits yield for APT measurements of Al, and the occurring strong Ga enrichment at grain boundaries is unbeneficial. The authors state that using cryogenic conditions avoids these common problems, which is interesting and surely a leap forward for site specific APT in Al alloys. Although the paper is well written, it would benefit from clearer visualization in some places: In Figure 1, it would be advantageous to show that there are two grains in some way. Compared to Figure 2, which demonstrates this (indexing the poles would be nice here), one may wonder why there is a low density area in one case and not in the other. Could the Ga segregation at GB’s also depended on the GB-type? These problems (which can relate to data visualization) should be improved and discussed more intensively.

minor issues:

Fig.1a: please add a scale bar.

Fig.2a: please add a scale bar.

Fig.2d and conclusion: Ga is not close to 0 %. There is still significant Ga level which is far above the natural Ga occurrence in the alloy 6016. It is in question why this level does not accumulate at this grain boundary.

Page 4: 80 K seems a bit high for Al-alloys. Typically 30 K or lower is used to measure Al alloys. This might effect composition and increases pole migration of Si. Please re-check the influence of this.

Fig.3: It could be enriched with further information (Tcryo, D, …)

6. PLOS authors have the option to publish the peer review history of their article (what does this mean?). If published, this will include your full peer review and any attached files.

Reviewer #1: No

Reviewer #2: No

---

## [Author Response · Author response to Decision Letter 0]

7 Feb 2020

The following can be found, along with figures, in the file "response to the reviewers". 

Reviewer #1:

In this paper two atom probe Al alloy samples are compared, one prepared with a FIB at room temperature and one prepared with the FIB cooled to cryogenic conditions. This is usually accompanied with a large effort to do, especially when investigating grain boundaries where Ga is present, which is known to lead to bad yield in the atom probe experiment. It is stated that Ga has a smaller diffusion length at lower temperatures, which is also underlined with a calculation of the diffusion coefficient. And concluded that no enrichment for Ga at the grain boundary in the cryo prepared sample was obtained. 

1. However, in general the paper would profit from a better representation, that in the room temperature FIB sample the second grain was already measured (as for Figure 2 b)). In general a larger concentration of Ga does not necessarily ensure the existence of a grain boundary. A visual representation of the grain boundary with an iso surface of the Ga distribution in the sample, or a Ga XY evaporation diagram, would be benefitial. Further the verification that the second grain was measured would be benefitial, either with showing loss of crystal information from grain to grain via SDMs or showing the different orientation with the XY evaporation diagram as in Figure 2 b), the reviewer can hardly see a difference in the shown ROI from grain 1 to grain 2 in Figure 1, apart from the low density region in between with increased Ga concentration, followed by a higher density region.

This subsection of the paper displaying results for the specimen prepared at room temperature mostly aims at confirming the low yield and fracture problems encountered in Al-alloys prepared by room-temperature Ga-FIB, and already reported several times in the literature, as was already mentioned in the paper. We agree that the quality of Figure 1, displaying the supposed grain boundary, is somewhat lower than that of Figure 2. This is because, as the reviewer stated, the success rate is very low for atom probe analyses of Al specimen including grain boundaries after Ga-FIB preparation at room temperature. 

To get this result, we analyzed a large number of specimens were prepared by site-specific lift-out at the grain boundary and early fracture of the specimens was systematically encountered. In comparison, all specimens prepared at cryogenic temperature yielded long datasets going through the grain boundary region. The specimen presented in Figure 1 is the only one that led to enough results that were indeed evidencing a large Ga increase. However, due to the fracture of the specimen soon after the supposed embrittled grain boundary, it was unfortunately impossible to get a XY evaporation histogram of good quality for the lower part. Still, a XY evaporation histogram for this part was added to the Figure 1. It is not as readable as the one of Grain 1, as fracture occurred However, some poles are visible. For an easier comparison, circles indicating the location of the poles in Grain 1 (upper grain) are reported on the XY evaporation histogram of Grain 2 (bottom grain), highlighting that there is no overlap, and therefore that the two domains have different crystallographic orientations.

New Figure 1:

Figure 1: Atom probe analysis of a grain boundary in a sample prepared at room temperature. 

(a) Part of the APT reconstruction of the specimen prepared by conventional FIB at room temperature. Atomic planes are visualized in grain 1 (top), and disappear in grain 2. Al atoms are represented as blue dots, and Ga atoms as red circles. (b) XY evaporation histogram of the detector showing crystallographic poles for grain 1. (c) XY evaporation histogram of the detector showing crystallographic poles for grain 2. The poles of grain 1 are reported on this histogram as red dashed circles for an easier readability. (c) Composition profile for Al and Ga across the grain boundary. Error bars are shown as lines filled with colour and correspond to the 2σ counting error.

The text was also modified as follows: 

“Detector maps are displayed for the upper part of the reconstruction in Figure 1b, and for the lower part in Figure 1c. For an easier readability, the poles observed for the upper part are reported on the lower evaporation histogram as red dashed circles. Although fracture of the specimen occurred soon after the Ga dense region, leading to an evaporation histogram of lower quality for the bottom part, some poles are still observed in Figure 1c. As highlighted by the red dashed circles, they do not overlap with the poles of Figure 1b. The upper and lower domains thus have different crystallographic orientation, and this information suggest that the dense Ga region corresponds to a grain boundary. This region also seems to have a lower density when looking at the Al in Figure 1a. This is likely related to the field evaporation behavior that is modified by the combination of a grain boundary (17) and the local presence of the high amount of Ga that seems to exhibit a low evaporation field and hence evaporated in a burst and led to significant distortions as can be observed for very low evaporation fields particles (18). Specimen failure in the vicinity or at the grain boundary itself occurred in several datasets, and can likely be attributed to Ga-induced embrittlement. A composition profile is plotted across the grain boundary (for the region displayed in Figure 1a) and is given in Figure 1c. It shows a clear Ga enrichment at the grain boundary, reaching values above 15 at.%, along with a depletion of Al. 

Such an accumulation of Ga has previously been reported to pertaining to microstructural features such as dislocations or grain boundaries (7,10,16). Here, our results suggest that the characterized feature corresponds to the grain boundary that we targeted during the preparation of the specimen. This explains the drastic planar Ga-enrichment. The fracture of the specimen soon after is also in line with embrittlement of grain boundaries caused by the Ga-indiffusion.” 

2. The behavior of Si at the boundary is described as increased at the boundary from almost 0% to 1.5 at.% the boundary, which is rather unexpected to the reviewer. Usually the Si content is in the order of 1 at.% in general in the alloy 6016. Further, Si is known to be seen to be present with a higher density in the reconstruction at the (111) pole. This pole is also seen on the XY evaporation diagram. In Figure 1 it is not indicated where the ROI was taken from, therefore if the (111) pole is presented, this could cause an increase of Si in the chosen ROI and potentially influence the evaluation of seen Si increase. However, the reviewer would rather expect that the boundary influences the behavior of Mg more, for both elements (Mg, Si) an evaluation as in Figure 1c would be good additional information. A scale for Figure 1a and Figure 2a would also be benefitial.

The authors double-checked the mentioned composition profile and apologize, as, indeed, the one presented was calculated close to a pole (since it was extracted from the displayed atomic reconstruction, which is done at a crystallographic pole), where a stronger density of Si was observed. This resulted in the observed segregation of Si, which would be unexpected as suggested by the reviewer. This issue related to the detection of Si at poles in Al is a common problem which can be related to surface diffusion during the field evaporation process (see B. Gault, F. Danoix, K. Hoummada, D. Mangelinck, H. Leitner, Impact of directional walk on atom probe microanalysis, Ultramicroscopy. 113 (2012) 182–191. doi:10.1016/j.ultramic.2011.06.005) or to species-specific trajectory aberrations that lead to a lower detection of Al-locally (see E.A. Marquis, F. Vurpillot, Chromatic aberrations in the field evaporation behavior of small precipitates., Microsc. Microanal. 14 (2008) 561–570. doi:10.1017/S1431927608080793). Several additional composition profiles were calculated at other locations across the boundary, without evidencing segregation of Si, as, for instance, the composition profiles plotted below (Ga is in red, Si in grey and Mg in purple):

Similar contents of Ga segregated at the boundary (ranging between 8 and 17 at% depending on the place of the measurement) were however quantified. This high variation is believed to be due to the fracture that occurred soon after the grain boundary, and the upper range of about 15 at.% was kept for the manuscript (that states “reaching values above 15 at.%”). In our opinion, it makes sense to remain consistent keep the composition profile calculated at location corresponding reconstruction, the plotted composition profile was not modified, but the sentence on the increase of Si was removed. 

Interestingly, no Mg segregation was observed, neither in the case of the specimen prepared at room temperature, nor in the case of the cryo-sharpened specimen.

Scale bars have been added. 

3. I cannot agree with the statement that "Ga is reduced close to 0 at.% (comparable to the level of background)" when comparing it to Figure 2d showing about 0.25 - 0.5 at.% Ga at average, which is clearly above usual background and in the order of the major alloying element concentrations (like Mg usually in the order of 0.4 at.% for 6016). Although Ga is present in industrial alloys, it only occurs in approx. two orders of magnitude less, which means that if there are Ga peaks present at this concentrations they are from FIB-implanted Ga.

Ga was detected in the specimen. Based on the isotopes found in the mass spectrum, it seems to originate from the milling step, as mostly one of the isotopes is detected. A figure is added in the supplementary materials showing the atomic reconstruction of the specimen, along with a composition profile of Ga showing a decrease from the surface of the tip to the bottom part. The mass spectrum (including a zoom in for the Ga peaks at 69 and 71 Da) is also provided:

Figure (supplementary materials): Ga implantation during milling for the cryo-prepared specimen.

(a) Atomic reconstruction and (b) composition profile for the Gallium showing a composition gradient from the top of the specimen to the bottom. (c) Corresponding mass spectrum of the dataset, that was cut at 100 Da (no peaks are observed at larger Da). The inset shows a zoom of the region between 64 and 74 Da, with the 69Ga+ peak and a much smaller 71Ga+ peak. The little peak at 65 Da corresponds to Cu, which is a classical impurity found in low concentrations. 

Based on these results, it is believed that the Ga detected on the composition profile of Figure 2d comes from the sample preparation, which does not impede with the main result, being the absence of Ga segregation at the grain boundary. 

The authors corrected the main text as follows: 

In the discussion part: 

“The Ga level measured there (0.25-0.5 at.%) is still higher than what can be expected for the commercial alloy. It is believed to originate from the sample preparation, as evidenced in the Figure 1 of the supplementary materials, which shows a contamination of Ga at the surface of the specimen, and a decrease of this content as the evaporation proceeds (see composition profile). A mass spectrum also shows that one isotope is predominantly obtained, reinforcing the hypothesis that the measured Ga comes from the FIB preparation. This does not impede the result that Ga does not segregate at the grain boundary.”

In the conclusion:

“Comparative APT experiments on a grain boundary of a commercial 6016 aluminum alloy show that the gallium composition at the grain boundary, above 15 at.% in the case of a room temperature Ga-FIB preparation, is reduced close to a very low level (fluctuating between 0.25 and 0.5 at.%) which does not increase at the grain boundary in the case of a cryo-FIB preparation.”

4. The specimen was transfered at room temperature to the atom probe, which thwarths somehow the argument that the cooling prevented Ga diffusion to the boundary / at the boundary. 

Here we must disagree with the reviewer. This is indeed a critical point: the room temperature transfer highlights the fact that cryo-transfer is not required on top of cryo-sharpening to obtain a specimen without Ga-contamination at the grain boundary. What matters most is the in-ward diffusion and associated accumulation of Ga during the preparation. 

We agree with the reviewer with the previous comment on the composition of Ga in the composition profile of Figure 2d, that is not exactly at 0 at.% for Ga, but it shows a very low composition and no increase specifically at the grain boundary, although the specimen was transferred at room temperature. The likely Ga contamination in the near-surface region likely lead to some Ga penetrating inside the specimen, but even though Ga diffused during the transfer, since the room-temperature would allow for such a diffusion, it did not reach a critical level. 

The following text was added to the discussion:

“The fact that the Ga composition does not increase specifically at the grain boundary, although the specimen was transferred at room temperature, suggests that cryo-transfer of the specimen is not required. Even in the present case where Ga contamination at the near-surface region occurred, and led to Ga diffusion in the specimen in the room-temperature transfer, it did not reach a critical level affecting the grain boundary.”

Some minor comments:

5. page 3: not "tension of 30 keV", "acceleration voltage of 30 keV"

This was modified.

6. experiments are examined at a rather "high" temperature of 80

80K is not an untypical temperature to perform APT. In this case, because of the brittle nature of grain boundaries in Al-alloys, in particular when prepared with a Ga-FIB, we tried to maximise the yield. In APT, this is typically achieved by lowering the magnitude of the electrostatic field, as achieved by a higher base temperature (see L. Yao et al., Optimisation of specimen temperature and pulse fraction in atom probe microscopy experiments on a microalloyed steel, Ultramicroscopu. 111 (2011) 648-651). The following sentence was added in the materials and methods section:

“The temperature for the analysis was chosen to maximize the yield (14).”

7. indicating the major poles of Figure 1b and Figure 2b,c would increase readability

The poles have been added.

8. There seems to be a lower density region where high amounts of Ga are present (Figure 1a), which is somehow misleadingly formulated at the main text.

The text was modified as follows:

“This information suggest that the dense Ga region corresponds to a grain boundary. This region also seems to have a lower density when looking at the Al in Figure 1a. This is likely related to the field evaporation behavior that is modified by the combination of a grain boundary (17) and the local presence of the high amount of Ga that seems to exhibit a low evaporation field and hence evaporated in a burst and led to significant distortions as can be observed for very low evaporation fields particles (18).”

Reviewer #2:

The work compares the preparation of APT samples using FIB at RT and cryogenic conditions. Ga induced via RT samples preparation often limits yield for APT measurements of Al, and the occurring strong Ga enrichment at grain boundaries is unbeneficial. The authors state that using cryogenic conditions avoids these common problems, which is interesting and surely a leap forward for site specific APT in Al alloys. Although the paper is well written, it would benefit from clearer visualization in some places: 

1. In Figure 1, it would be advantageous to show that there are two grains in some way. Compared to Figure 2, which demonstrates this (indexing the poles would be nice here), one may wonder why there is a low density area in one case and not in the other. Could the Ga segregation at GB’s also depended on the GB-type? These problems (which can relate to data visualization) should be improved and discussed more intensively.

This question was already addressed in the answer to the reviewer 1’s first point, and the Figure 1 was modified, by adding the histogram corresponding to Grain 2. The difference of segregation of Ga at the various types of GB is also briefly discussed in the paper, mostly at the beginning of the discussion. 

minor issues:

2. Fig.1a: please add a scale bar.

A scale bar was added.

3. Fig.2a: please add a scale bar.

A scale bar was added.

4. Fig.2d and conclusion: Ga is not close to 0 %. There is still significant Ga level which is far above the natural Ga occurrence in the alloy 6016. It is in question why this level does not accumulate at this grain boundary.

This topic has also been discussed in the answer to the Reviewer 1. 

5. Page 4: 80 K seems a bit high for Al-alloys. Typically 30 K or lower is used to measure Al alloys. This might effect composition and increases pole migration of Si. Please re-check the influence of this.

This point was also raised by the Reviewer 1 and an answer was provided above. 

6. Fig.3: It could be enriched with further information (Tcryo, D, …)

The temperatures were added to the Figure. Regarding the other information; they are actually not quantified, as the diffusion coefficients calculated only provide values to compare the two cases, but do not actually allow quantifying an affected distance.

---

## [Decision Letter · Decision Letter 1]

11 Mar 2020

PONE-D-19-33571R1

New approach for FIB-preparation of atom probe specimens for aluminum alloys

PLOS ONE

Dear Dr Lilensten,

Thank you for submitting your manuscript to PLOS ONE. I have completed the review of your manuscript and a summary is appended below. The reviewers recommend reconsideration of your paper following Minor Revision. I invite you to resubmit your manuscript after addressing all reviewer comments.

Reviewer #1

The authors have addressed the raised points, besides some minor points, to satisfaction of the reviewer.

Minor points:

- Something went wrong in labeling the poles, please check and correct Figure 1 b) and Figure 2 b).

- There is a display error in the bibliography, reference [25]. "$\\ensuremath\\Sigma5$"

Reviewer #2

The authors have addressed my points and the paper can be published after minor correction.

I am not fully convinced by Fig. 1. For me the difference between 1b and 1c is not totally clear. This could also be the same orientation. Maybe the authors can add a comment.

We would appreciate receiving your revised manuscript by Apr 25 2020 11:59PM. To enhance the reproducibility of your results, we recommend that if applicable you deposit your laboratory protocols in protocols.io, where a protocol can be assigned its own identifier (DOI) such that it can be cited independently in the future. For instructions see: http://journals.plos.org/plosone/s/submission-guidelines#loc-laboratory-protocols

We look forward to receiving your revised manuscript.

Kind regards,

Hamid Reza Bakhsheshi-Rad

Academic Editor

PLOS ONE

Reviewers' comments:

Reviewer's Responses to Questions

**Comments to the Author**

1. If the authors have adequately addressed your comments raised in a previous round of review and you feel that this manuscript is now acceptable for publication, you may indicate that here to bypass the “Comments to the Author” section, enter your conflict of interest statement in the “Confidential to Editor” section, and submit your "Accept" recommendation.

Reviewer #1: (No Response)

Reviewer #2: All comments have been addressed

2. Is the manuscript technically sound, and do the data support the conclusions?

Reviewer #1: Yes

Reviewer #2: Yes

3. Has the statistical analysis been performed appropriately and rigorously? 

Reviewer #1: Yes

Reviewer #2: Yes

4. Have the authors made all data underlying the findings in their manuscript fully available?

Reviewer #1: Yes

Reviewer #2: Yes

5. Is the manuscript presented in an intelligible fashion and written in standard English?

Reviewer #1: Yes

Reviewer #2: Yes

6. Review Comments to the Author

Reviewer #1: The authors have addressed the raised points, besides some minor points, to satisfaction of the reviewer.

Minor points:

- Something went wrong in labeling the poles, please check and correct Figure 1 b) and Figure 2 b).

- There is a display error in the bibliography, reference [25]. "$\\ensuremath\\Sigma5$"

Reviewer #2: The authors have addressed my points and the paper can be published after minor correction.

I am not fully convinced by Fig. 1. For me the difference between 1b and 1c is not totally clear. This could also be the same orientation. Maybe the authors can add a comment.

7. PLOS authors have the option to publish the peer review history of their article (what does this mean?). If published, this will include your full peer review and any attached files.

Reviewer #1: No

Reviewer #2: No

---

## [Author Response · Author response to Decision Letter 1]

13 Mar 2020

Reviewer #1:

The authors have addressed the raised points, besides some minor points, to satisfaction of the reviewer.

Minor points:

- Something went wrong in labeling the poles, please check and correct Figure 1 b) and Figure 2 b).

The two figures have been double checked and, indeed, corrected.

- There is a display error in the bibliography, reference [25]. "$\\ensuremath\\Sigma5$"

It has been corrected. 

Reviewer #2:

The authors have addressed my points and the paper can be published after minor correction.

I am not fully convinced by Fig. 1. For me the difference between 1b and 1c is not totally clear. This could also be the same orientation. Maybe the authors can add a comment. 

The text was modified as follows:

“As highlighted by the red dashed circles, it appears that there is a shift of the position of the main poles in Fig 1b. This is an indication that the upper and lower parts, on either side of the Ga-rich region, have different crystallographic orientation, and this information suggest that the dense Ga region corresponds to a grain boundary.”

---

## [Editor Report · Decision Letter 2]

18 Mar 2020

New approach for FIB-preparation of atom probe specimens for aluminum alloys

PONE-D-19-33571R2

Dear Dr. Lilensten,

We are pleased to inform you that your manuscript has been judged scientifically suitable for publication and will be formally accepted for publication once it complies with all outstanding technical requirements.

With kind regards,

Hamid Reza Bakhsheshi-Rad

Academic Editor

PLOS ONE
---

## [Editor Report · Acceptance letter]

20 Mar 2020

PONE-D-19-33571R2 

New approach for FIB-preparation of atom probe specimens for aluminum alloys 

Dear Dr. Lilensten:

I am pleased to inform you that your manuscript has been deemed suitable for publication in PLOS ONE. Congratulations! Your manuscript is now with our production department. 

With kind regards,

on behalf of

Dr. Hamid Reza Bakhsheshi-Rad 

Academic Editor

PLOS ONE